# NOMA-Based VLC Systems: A Comprehensive Review

**DOI:** 10.3390/s23062960

**Published:** 2023-03-09

**Authors:** Syed Agha Hassnain Mohsan, Muhammad Sadiq, Yanlong Li, Alexey V. Shvetsov, Svetlana V. Shvetsova, Muhammad Shafiq

**Affiliations:** 1Optical Communications Laboratory, Ocean College, Zhejiang University, Zheda Road 1, Zhoushan 316021, China; 2Shenzhen Institute of Information Technology, Shenzhen 518172, China; 3Ministry of Education Key Laboratory of Cognitive Radio and Information Processing, Guilin University of Electronic Technology, Guilin 541004, China; 4Department of Smart Technologies, Moscow Polytechnic University, 107023 Moscow, Russia; 5Faculty of Road Transport, North-Eastern Federal University, 107023 Yakutsk, Russia; 6School of Computer, 107023 Khabarovsk, Russia; 7Cyberspace Institute of Advanced Technology, Guangzhou University, Guangzhou 510006, China

**Keywords:** NOMA, successive interference cancellation (SIC), spectral efficiency, massive connectivity, multiple access, visible light communication, resource allocation

## Abstract

The enhanced proliferation of connected entities needs a deployment of innovative technologies for the next generation wireless networks. One of the critical concerns, however, is the spectrum scarcity, due to the unprecedented broadcast penetration rate nowadays. Based on this, visible light communication (VLC) has recently emerged as a viable solution to secure high-speed communications. VLC, a high data rate communication technology, has proven its stature as a promising complementary to its radio frequency (RF) counterpart. VLC is a cost-effective, energy-efficient, and secure technology that exploits the current infrastructure, specifically within indoor and underwater environments. Yet, despite their appealing capabilities, VLC systems face several limitations which constraint their potentials such as LED’s limited bandwidth, dimming, flickering, line-of-sight (LOS) requirement, impact of harsh weather conditions, noise, interference, shadowing, transceiver alignment, signal decoding complexity, and mobility issue. Consequently, non-orthogonal multiple access (NOMA) has been considered an effective technique to circumvent these shortcomings. The NOMA scheme has emerged as a revolutionary paradigm to address the shortcomings of VLC systems. The potentials of NOMA are to increase the number of users, system’s capacity, massive connectivity, and enhance the spectrum and energy efficiency in future communication scenarios. Motivated by this, the presented study offers an overview of NOMA-based VLC systems. This article provides a broad scope of existing research activities of NOMA-based VLC systems. This article aims to provide firsthand knowledge of the prominence of NOMA and VLC and surveys several NOMA-enabled VLC systems. We briefly highlight the potential and capabilities of NOMA-based VLC systems. In addition, we outline the integration of such systems with several emerging technologies such as intelligent reflecting surfaces (IRS), orthogonal frequency division multiplexing (OFDM), multiple-input and multiple-output (MIMO) and unmanned aerial vehicles (UAVs). Furthermore, we focus on NOMA-based hybrid RF/VLC networks and discuss the role of machine learning (ML) tools and physical layer security (PLS) in this domain. In addition, this study also highlights diverse and significant technical hindrances prevailing in NOMA-based VLC systems. We highlight future research directions, along with provided insights that are envisioned to be helpful towards the effective practical deployment of such systems. In a nutshell, this review highlights the existing and ongoing research activities for NOMA-based VLC systems, which will provide sufficient guidelines for research communities working in this domain and it will pave the way for successful deployment of these systems.

## 1. Introduction

As the need for wireless data transmission continues to skyrocket, future wireless networks are going to require high spectrum efficiency, high bandwidth, and huge interconnectivity. The existing radio frequency (RF) band is getting increasingly overcrowded, making it difficult to meet these demands [1]. Because of its greater bandwidth, which does not compromise the RF spectrum, its energy efficiency, and its capacity to offer ubiquitous connection, visible light communication (VLC) has recently been suggested as a credible option for indoor wireless communication [2]. VLC can meet the above-mentioned objectives for future wireless networks by deploying an efficient multiple-access method [3]. Non-orthogonal multiple access (NOMA), which has great spectrum efficiency, is among the most transformative multiple-access techniques that have recently been suggested. Multiple users might share the same frequency/time resource block in NOMA, allowing for huge connections and great throughput. On the transmitting end, NOMA uses multi-user superposition transmission (MUST), while it uses multi-user detection (MUD) at the receiver for detection purposes, and decoding is performed through successive interference cancellation (SIC) [4].

There are several reports in scholarly articles on the use of NOMA techniques in VLC. Multiple factors make NOMA an excellent choice for downlink VLC systems, as stated in [5,6]. Firstly, a VLC cell only needs to accommodate a modest number of users for NOMA to function properly. Secondly, since the channel is relatively constant in VLC systems, channel estimation is simpler. Therefore, NOMA can execute the load distribution at the transmitting end and the interference cancellation at the receiving end by making use of the channel status information. Higher-order modulation methods, including quadrature amplitude modulation (QAM), can be used to further enhance the spectrum efficiency of NOMA-based VLC systems. QAM cannot be implemented directly to VLC due to the complex-valued and bipolar symbols it generates, which are necessary for indoor VLC, which is dependent on intensity modulation/direct detection (IM/DD). To get over this problem, we might employ O-OFDM technology, which uses the idea of Hermitian symmetry to produce a signal in the time domain with real values. If the OFDM signal is amplified with a positive direct current (DC), like in the direct-current O-OFDM (DCO-OFDM), or if we restrict the transmission to odd subcarriers alone, as in the irregularly clipped O-OFDM (ACO-OFDM), a unipolar time-domain signal is generated [7]. The trade-off for NOMA systems’ ability to achieve great spectral efficiency and high connectivity is an enhancement in data detection error caused by inevitable inter-user interference [8]. Because of the potential for increased inter-user interference when using NOMA with higher cognitive modulation, cautious power allocation (PA) is required. In a recent study [9], the authors determine the PA constraint for enabling NOMA with higher-order modulation. Even though NOMA systems have indeed been studied extensively, some NOMA-based VLC research focuses on lower-order modulation techniques and assumes perfect SIC. Consequently, it is essential to properly address the open problem of how NOMA-aided VLC systems using higher-order modulation handle errors.

Generally, the NOMA approaches were discussed in the literature to boost the throughput and reliability of VLC systems. Unlike OMA approaches, NOMA enables numerous users to use the same frequency/time resource blocks at the loss of certain IUI, resulting in effective resource usage [10]. Downlink PD-NOMA, in particular, depends on the SC idea at the transmission end for multiplexing distinct users’ data streams in the power domain and on the SIC approach at the end users to decipher receiving data [11]. NOMA works by providing users with varying power levels that are usually related to respective channel gains. To decrypt the data, the strongest users initially utilize SIC for decoding the data of weak users, remove it from associative receivers, and then decrypt their data, but the weak users decrypt their data immediately and face IUI caused by the stronger users’ data superposition. Despite the numerous advantages that VLC provides, it has a number of flaws that keep present technology from meeting the expectations of 6G networks.

The first constraint is the minimal interaction range caused by the short wavelengths of visible light signals. It leads to significant propagation losses since the VLC channel strength deteriorates substantially as the space between receiving and transmitting devices grows, besides the fact that emission spectra are easily obstructed by obstructions [12]. Furthermore, unlike its typical RF counterpart, the VLC channel is not isotropic, which means that the geometry of the receiving and transmitting instruments have a substantial impact on channel gains [13]. Consequently, the VLC channel quality varies, and the efficiency of sophisticated multiple access strategies, for instance, NOMA, suffers when integrated with VLC systems. An overview of downlink NOMA-VLC for two users is presented in Figure 1. We have summarized recent works on NOMA in Table 1.

### 1.1. Background

VLC is not intended to substitute for RF, but instead to supplement it; VLC networks may be incorporated into 5G wireless communication systems by connecting to existing optical fiber networks [26]. Whether you have been trying to get a message indoors, outdoors, underwater, or in underground settings, VLC can be an integral part of it. In order to offer both light and communication, VLC proves to be an energy-efficient technology that makes use of the widespread use of light-emitting diodes (LEDs) [27]. The traditional VLC system utilizes basic, commercially available sources, such as a light-emitting diode (LED) for transmission and a photodetector (PD) for reception [26]. Intensity modulation (IM) allows LEDs to send data by subtly adjusting the brightness of the light at extremely fast rates [28]. Direct detection (DD) involves generating an electric current at the receiver end that is proportional to the fluctuation in optical power of the receiving end [29]. This is done with photodetectors or image sensors. The development of VLC technology is still in its infancy, and it will take significant work to see it extensively used in real-world settings. From undersea communication to military reasons, as well as broadband access, interior localization, power line communication (PLC), and intelligent transportation systems (ITS) [30], many different applications are expected to be widely implemented in the next few years.

Despite these advantages, however, VLC systems still have several drawbacks that must be fixed before their full potential can be realized [28]. Technically, the low modulation bandwidths of LEDs (a few MHz) are the primary obstacle to constructing VLC systems with large attainable data speeds. Blue LEDs containing yellow phosphorous and RGB LEDs with a modulating bandwidth of 2–5 MHz and 10–20 MHz are frequently utilized for VLC due to their white light [31]. When compared to RGB LED, devices that rely on blue LEDs offer more cost-effectiveness and energy efficiency; hence, they are gaining popularity in lighting systems.

However, RGB’s greater modulator bandwidth empowers it highly suitably for transmission. Nevertheless, to solve the modulation bandwidth challenge and employ VLC systems to their maximum potential, economic improvement of high-order modulation methods, MIMO, frequency reuse, and sophisticated channel access techniques are recommended [32]. 

#### 1.1.1. NOMA for VLC

There are a number of impressive advantages to VLC, but it also has some serious drawbacks. In particular, the relatively limited modulation bandwidth of LED is a critical concern when it comes to the design of high-data-rate VLC systems. While VLC enables numerous users to access the network at once, conventional OMA techniques do not promote efficient resource usage. As a result, NOMA, and more specifically, PD- NOMA, can be viewed as a suitable multiple access strategy that provides adequate bandwidth for indoor VLC systems [33]. NOMA allows for significant increases in throughput and other performance metrics. Consequently, NOMA may be conceived of as an efficient and powerful multiple-access strategy for enhancing the spectrum efficiency of VLC systems. Not only can NOMA efficiently superimpose a small number of users, but VLC systems, which employ LEDs as transmitters, are also in line with this capability. In closed spaces, the LED sources act as access points for a limited number of users. In order to facilitate user power allocation, demultiplexing, and decoding order, the SIC process of NOMA necessitates CSI at both the transmitter and receiver sides. Due to the relatively constant channel in VLC systems, this issue does not arise as often. Due to the close proximity of the LED and PD in VLC networks, NOMA operates more efficiently under high SNR conditions. Channel gain disparities between users in VLC systems can be minimized by tweaking the LEDs’ transmission angles and the PDs’ FOVs. The best multiple-access technique for very low latency (VLC) systems is non-orthogonal multiple access (NOMA), since it performs well even when user channels have large disparities in gain [34]. Figure 2a presents a basic VLC system while Figure 2b provides a conceptual diagram of a NOMA-VLC system. In Figure 2b, a transmitter (an LED) sends data to two receivers. When compared to an instance where the two users are far apart, the signal intensity for user 1 is greater than when the line of sight to the LED is shorter, and vice versa. The NOMA technique determines that user 2 will be given a more powerful h2 and will be able to directly decode its message signal. With regard to h1, on the other hand, it will get a lower power request and will have to go through the SIC phenomenon to decode its required signal [18]. Without any doubt, NOMA schemes offer better performance for VLC systems than a current OMA scheme. Continuously implementing NOMA on all users cannot be feasible because of increase in SIC computational complexity due to increasing the number of users. In order to tackle this challenge, user pairing is adopted as the alternative solution to decrease the SIC decoding complexity. In addition, a compatible power allocation and user pairing can play a viable role in improving the performance of NOMA-VLC systems [29]. Since NOMA is a promising candidate for next generation cellular networks, its feasibility in VLC is a subject of high research interest. Here, we have discussed some benefits of using hybrid RF/VLC NOMA and cooperative NOMA for VLC systems: In NOMA-VLC systems, a cooperative NOMA scheme can be incorporated; however, the strong user may not intend to utilize some power to forward the signal to the weak user. Thus, it is possible to investigate the approaches where the strong user can harvest energy from the optical sources and then it can consume it to forward signal to the weak user.When a VLC system is based on several access points (APs) for the transmission of power and information to multiple users, if the number of APs is lower than users, then a critical issue is to adopt OMA or NOMA for energy harvesting and user scheduling. As NOMA offers higher data rates compared to OMA, the VLC-NOMA system is thus adopted to attain the required data rates for all users with a small amount of transmit power.Cooperative NOMA is basically introduced in RF networks for the exploitation of redundant data in NOMA systems, and for the compensation of weak user facing co-channel interference. The cooperative NOMA can be integrated in VLC systems or through relaying systems. While assuming a VLC system of two users, all users can collect data from RF or VLC networks simultaneously. In such cases, the strong user can decode the weak user’s signal and forward it to the intended user through Bluetooth or Wi-Fi. The weak user can then mix the VLC and RF signals through combining methods.In the literature, several studies have reported modulation and coding schemes for RF-NOMA systems. Since the modulation and coding schemes for NOMA-VLC are different from RF-NOMA, finding the novel modulation and coding schemes for NOMA-VLC systems is worthy for successful deployment of these systems.As VLC systems are prone to SNR fluctuation, some users receive poor QoS due to handover overhead, inter-cell interference, and LoS blockage, while other users get a high QoS. A cooperative NOMA-based scheme can be adopted where good-serviced users can support weak users through RF links. Thus, a hybrid system can be easily established to provide good service for all users.

#### 1.1.2. NOMA for RF 

Multiple access strategies in VLC systems seem to be similar to those employed in RF systems [28]. Various orthogonal and non-orthogonal RF multiple access approaches have successfully implemented network resource sharing by considering a large number of users. To accommodate a large number of users in 1G cellular networks, frequency division multiple access (FDMA) was considered to divide the allocated frequency bandwidth among users. On the contrary, 2G cellular technologies are founded on time division multiple access (TDMA), which permits several users to utilize the same frequency range by splitting it into time periods [3]. Similarly, 3G cellular technology employed the code division multiple access (CDMA) approaches, which permitted various users to be assigned time intervals and frequencies with a unique code to avoid interference. Various frequency subcarriers are given to various users in fourth-generation (4G) networks employing orthogonal frequency division multiple access (OFDMA) [35].

It should be kept into account that various users are assigned to orthogonal resources in the aforesaid OMA strategies (time, frequency, and code). Nonetheless, 5G and B5G networks must offer services to lots of consumers who are using high data rates [32,36]. Non-orthogonal multiple access (NOMA), in comparison to OMA, enabled numerous users to share the same frequency range at once, thereby significantly boosting the spectral efficiency of the system. NOMA empowers the simultaneous transmission of multiple users considering the same degree of freedom (DoF) through superposition coding with various power levels. It employs multiple user detection (MUD) to separate the users sharing the same DoF, as illustrated in Figure 3. Meanwhile, through exploiting the received power disparity, advanced signal processing methods, such as SIC, can be used to retrieve the intended signals at the receiver. NOMA can substantially enhance sum rate and spectral efficiency as compared to OMA. In addition, NOMA has the potential to enlarge the number of users by ensuring controllable symbol collision in the same DoF. Therefore, it ensures high overloading transmission and further enhances the system capacity given limited resource (antennas or spectrum) [37]. NOMA is made up of two primary types: code domain NOMA (CD-NOMA) and PD-NOMA [38,39]. The fundamental working principle of CD-NOMA is the same as CDMA, allowing many users to share all available resources (time, frequency). However, CD-NOMA benefits more users by making use of sparse dispersion or non-orthogonal low cross-correlation sequences [40]. Low-density spreading CDMA (LDS-CDMA), sparse code multiple access (SCMA), and LDS employing OFDM are only a few of the classes that make up CD-NOMA (LDS-OFDM). For more information on CD-NOMA, concerned readers are directed to [41,42]. Spatial division multiple access (SDMA), building block sparse-constellation-based OMA (BOMA), lattice partition multiple access (LPMA), and pattern division multiple access (PDMA), are other NOMA algorithms that are used in many fields [43]. A viable contender for surpassing 5G networks, whereby numerous users broadcast and receive the full available frequency and temporal resources utilizing varying power levels, is the extensively used PD-NOMA-aided system, which is comparably less sophisticated than the CD-NOMA-aided system [44]. Because developing RF and VLC techniques has the ability to ensure high system throughput and widespread connectivity, some studies [8,45] concentrate on PD-NOMA as an appealing method. PD-NOMA exploits the power domain to adjust several users in the same DoF and operates SIC at users with better channel conditions. Some studies consider a hybrid NOMA/OMA technique based on accurately developed precoders and decoders, thus empowering a more effective use of the available DoFs as compared to conventional OMA and NOMA [46]. Most of the existing studies on RF-NOMA mainly address the issue of quality of service (QoS) requirements [22,23], while only a few focus on improving the energy efficiency of RF-NOMA [21]. In [20], the authors introduced a distance-dependent PA (DDPA) method for a NOMA system for mMIMO, carried our performance analysis, and compared it with static power allocation (SPA) methods which are commonly reported in the literature for RF-NOMA systems. They stated that SPA outperforms DDPA in the context of energy efficiency and overall sum rate. Even though VLC-NOMA systems are inherently more energy-efficient compared to RF-NOMA systems, their energy efficiency can be effectively enhanced by proper power allocation. Some recent works have also focused on enhancing the energy efficiency of RF-NOMA systems [15]. In Table 2, we have presented a comparison of different multiple access techniques for wireless cellular networks.

### 1.2. Scope and Contributions

Readers interested in reading NOMA-based VLC systems should read this review. This paper offers an in-depth analysis of existing research works on NOMA-based VLC systems. The purpose of this review is to assist readers with a thorough understanding of the NOMA-based VLC systems from a variety of perspectives. It provides a comprehensive discussion of MISO/MIMO techniques for NOMA-based VLC systems. It also explains the integration of emerging technologies such as IRS, OFDM, MIMO, UVLC, PLS, ML, etc. into NOMA-based VLC. Finally, it explains a number of potential challenges and future research areas for further breakthrough novelties.

### 1.3. Organization of the Paper

This work is arranged as follows. Section 2 discusses the associated NOMA-based VLC research contributions. Section 3 discusses the integration of NOMA-based VLC with emerging technologies. Section 4 highlights several challenges and open research issues. Section 5 concentrates on future research directions. Lastly, Section 6 brings the paper to a close. 

## 2. Related Work

In this section, we review the studies on NOMA-VLC systems previously reported in the literature. We highlight several works which have been performed on the NOMA-VLC. Recently, NOMA has emerged as a promising candidate to support 5G technologies. All NOMA users can get support for frequency and time resources which are advantageous to enhance the sum rate of NOMA-based VLC systems [47]. NOMA has proven to be more appropriate for downlink VLC systems [48]. Similarly, some studies have developed a downlink NOMA-VLC system using LED source and multiple users [49]. 

Previous publications [5,49] suggested and demonstrated studies on NOMA implementation in VLC. In [5], a PA approach termed “gain ratio power allocation” (GRPA) that takes into account channel characteristics in NOMA-VLC systems was presented, which improves system performance over static power allocation (SPA). The researchers also discovered that by altering the field of view (FOV) of photodiodes (PDs) and the semi-angle of transmitting LEDs, the total rate may be increased even further. Based on actual indoor VLC channel models, the investigators in [34] demonstrated the advantages of NOMA in VLC over OFDMA. Ref. [50] researched and assessed the sum rate of NOMA-based VLC system considering one LED and several users. In parallel, an alternative power allocation approach that enhances system capacity while considering illumination intensity and user fairness was developed in [51], and a low-complexity power control algorithm was also provided and assessed. The utilization of NOMA paired with OFDMA in VLC systems was proven in [49] to reach increased capacity via trials. Nevertheless, the aforesaid research into system performance upgrades is insufficiently thorough since it only includes a single or partial component that influences performance of the system. In reality, while prior studies have increased the sum rate, there is a notable gap of tradeoff between sum-rate efficiency and fairness, user quality of service (QoS), and optical intensity at the same time. Furthermore, most recent researchers emphasize on the situation of just two users, ignoring the scenario of inadequate SIC, wherein non-negligible leftover interference occurs owing to user movement and feedback delays at the receiver, resulting in poor detection accuracy and a fall in the attainable sum rate.

In a recent work [47], the authors set up a single-LED, N-user NOMA-VLC framework within. In addition, this work represents the first effort to create an optimum power allocation technique (also known as MFOPA) by taking into account several elements that govern the system’s performance concurrently. The authors’ mission is to optimize system throughput while simultaneously satisfying the needs of all of customers with regard to the QoS, equity, and the safety of their data. In addition, the interference cancellation factor must take into consideration the interference that remains after the SIC procedure has been applied at the receiver, which is also considered in the MFOPA plan that has been presented. For a variety of system sum rate performance metrics, numerical findings demonstrate that MFOPA can substantially improve the system. Because of this, the suggested MFOPA technique may be tested and proven to be both effective and reliable.

In [5], the developers of the NOMA-based VLC suggested using the gain ratio power allocation (GRPA) technique. Light fidelity (LiFi) networks are fully-networked VLC systems, and the researchers in [52] discovered that NOMA can often boost throughput in these networks. Additionally, they discovered that NOMA might enhance attocell edge user performance without significantly impairing that of other users. For a system similar to a 2 × 2 MIMO-NOMA-based VLC system, the experts in [17] suggested a normalized gain difference power allocation (NGDPA) approach and showed that it beats GRPA in the context of the system sum rate. It has been suggested in [53] to employ improved PA for OFDM-NOMA-aided VLC with any number of multiplexed users. Both subcarrier and user levels of power optimization have been carried out. The findings showed that the system sum rate was improved as opposed to the GRPA method. Closed-form formulas of the system ergodic sum rate and coverage probability were developed by the authors in [34]. Additionally, the likelihood that each individual rate in NOMA is higher compared to OMA has been calculated. The researchers in [54] suggested a method for data detection called the ergodicity and comparison (EAC) method in order to mitigate the SIC process and enhance the performance of the NOMA-aided VLC system through M-QAM. A constellation partitioning coding (CPC)-free NOMA approach for downlink VLC systems has been suggested in [48], and it makes use of the ideas of uneven constellation demapping and extensible SIC-free NOMA (UCD). It is possible to effectively reduce the impact of the error propagation that occurs from insufficient SIC. For a NOMA-aided VLC system, the authors of [55] suggested symmetric superposition coding (SSC) and symmetric SIC (SSIC) decoding. Many of the nearby second-user decision areas using SSC and SSIC will display the same symbol. As a result, the impact of error propagation is reduced. Even though this study indicated improvements in the system’s performance, it failed to pinpoint the PA that ensures optimal performance. Under the conditions of perfect CSI, a closed-form formula for the BER of a single LED downlink NOMA-aided VLC system using on-off keying (OOK) has been developed in [56]. In addition, the upper bound under noisy and out-of-date CSI has a straightforward and precise estimation. A numerical equation for the SER of a multi-LED downlink NOMA-aided VLC system was obtained by the authors in [57]. Using Nakagami-m fading channels, the pairwise error probability (PEP) of NOMA systems was examined in [58]. The bit error rate’s precise union bound is obtained using PEP expressions (BER). Ref. [59] looked at how well downlink NOMA systems performed while operating over Nakagami-m fading channels. They looked at the two-user and three-user NOMA situations. The calculated BER equations in [60,61] are for the quadrature phase-shift keying (QPSK) technique. A recent work [9] focuses on the performance of downlink NOMA-aided VLC systems using higher-order modulation methods. In a more generic case, square M-QAM is used to create the user signals. It is also possible to apply the analyses presented in this study to other NOMA-based communication networks. Table 3 summarizes recent works on NOMA-based VLC systems.

## 3. Integration of NOMA-Based VLC with Emerging Technologies

In this section, we present a brief description of the NOMA-VLC systems integrated with different emerging technologies. We mainly focus on the benefits of this coexistence with these technologies as discussed in the literature. 

### 3.1. MISO/MIMO Techniques in NOMA-Based VLC Systems

By utilizing illuminating LED arrays, MIMO has been widely used in VLC systems as a natural and effective technique to widen system coverage and boost system capacity [68,69]. By installing multiple sources at the transmitter or receiver, additional spatial DoFs can be enabled. Massive MIMO (mMIMO) can also substantially boost the spectral efficiency due to the large number of spatial DoFs introduced by the multiple sources [70]. However, considering the future demand of coverage for an excessive number of users, the spatial DoFs provided by MIMO or mMIMO are not sufficient. In this case, NOMA can be incorporated to serve a higher number of users. The use of NOMA in MIMO-VLC systems has not received much research. The experimental verification of a MIMO-NOMA-aided VLC system in [71] did not take power allocation into account. In MIMO-NOMA-aided VLC systems, the PA strategies of single-LED NOMA-VLC systems cannot be immediately applied. For MIMO-NOMA radio-frequency (RF) systems, a number of PA techniques, including signal alignment [72], hybrid precoding, and post-detection [73] have been discussed in the literature thus far. These techniques, however, have a significant computational cost. For the possible widespread use of the MIMO-NOMA methodology in real-world VLC systems, effective PA techniques with low computing complexity are crucial.

In a recent study [17], the authors extend NOMA to MIMO-VLC systems and provide a cutting-edge power distribution technique, called normalized gain difference power allocation (NGDPA), for effective and simple power distribution in MIMO-NOMA-VLC systems. Numerical simulations are used to assess the total rate performance of a 2 × 2 MIMO-NOMAVLC system installed inside. It is demonstrated that, as compared to NOMA with GRPA, the achievable sum rate of the 2 × 2 MIMO-VLC system may be substantially increased by using NOMA with the suggested NGDPA approach. A 2 × 2 MIMO-VLC system for k number of users is illustrated in Figure 4.

NOMA is used in a MIMO VLC system with many users in [74]. The authors of these studies demonstrate that a certain type of NOMA power allocation—normalized gain difference power allocation (NGDPA)—greatly improves the achievable sum rate. The BER performance of OQAM-OFDM-aided MIMO-NOMA over VLC has been evaluated as a function of the power allocation ratio between users [75]. Researchers point out that a MIMO-NOMA transmission has not thus far been considered in the articles on multiuser MIMO VLC systems described in the studies in order to mitigate the performance loss of multiuser precoding techniques occurred by the strong correlation between channel gain vectors of various users. VLC systems and MIMO-VLC systems [71,76] have explored PD-NOMA as well. While power domain NOMA has the potential to improve performance, it highly depends on power allocation algorithms and the user pairing/grouping techniques that are used [65], making it challenging to attain optimal performance in real-world cost-efficient MIMO-VLC systems.

#### 3.1.1. VLC-NOMA for Underwater Applications

VLC is an emerging tool with high flexibility, huge capacity, a cheap cost, and no licensing requirements that have recently been examined for underwater applications [77]. When compared to acoustic technology, underwater visible light communication (UVLC) using blue/green light sources may efficiently enable low-latency and high-speed transmission [78]. UVLC, as opposed to underwater wireless electromagnetic communication, offers the enticing features of having no electromagnetic radiation and strong anti-interference capabilities [79]. Nevertheless, because of the significant absorption and scattering caused by water, the UVLC system still has a restricted transmission rate and communication distance. Some potential methods for improving UVLC system performance include enhanced coding techniques, updated modulation techniques, and MIMO transmission schemes. UVLC features improve optical detectors, including single photon avalanche diodes (SPAD), to increase communication distance, enabling improved detection capability [80]. 

The effectiveness of the NOMA-VLC system has been studied with respect to LED half-power semi-angles, photodiode fields of view (FOVs), power allocation coefficients, and channel conditions [81]. In [81], the authors offer a NOMA-UVLC system that relies on the PDM with several color LED sources introduced to increase system performance. Initially, the UVLC channel’s attenuation properties were explored and discussed. On this premise, a NOMA-UVLC system with unique data on distinct carriers is created and constructed with the goal of improving total transmission rate and spectrum efficiency. When the system uses two similar or distinct light sources, the BER curve of the NOMA-UVLC system varies in accordance to the transmission rates. Furthermore, the authors carried out a comparison of the experimental findings between the underwater and the air environment.

Underwater optical wireless communication can also benefit from NOMA technology by significantly increasing transmission capacity and lowering user interference [82,83]. The practicality of the suggested system model was confirmed by an examination of NOMA’s performance in a variety of underwater circumstances, as presented in [84]. A NOMA-UVLC system with a photon-counting receiver was presented in ref. [85], which demonstrated excellent sensitivity to weak signals and BER performance at extremely low signal strengths. However, the emphasis of these efforts was purely theoretical and/or simulated. There is also evidence that water strongly absorbs light, with the exception of the blue-green portion of the visible light spectrum. Therefore, various LED light sources at different wavelengths experience variable levels of power attenuation in water, providing a theoretical basis for the incorporation of PDM technology into UVLC systems in order to increase channel capacity.

#### 3.1.2. PD-NOMA for Underwater Applications 

Because of the rising usage of underwater sensor networks (USNs) for a variety of tasks, including environmental assessment, port security, oil exploration, tactical surveillance, and collecting data, scientists have begun to examine underwater wireless networking options [86,87]. Acoustic communications have been a popular approach for USNs since they can sustain transmission lengths of several kilometers with modest data rates on the order of kbps. UVLC has been offered as a supplementary connection option, with data speeds in the tens of megabits per second (Mbps). Light is significantly attenuated when it travels through water, especially at ultraviolet and infrared wavelengths. The optimal wavelength for underwater transmission is in the blue-green region of the VL spectrum. Whereas the green spectrum attenuates less in seawater, the blue spectrum attenuates more in the wide ocean [87]. 

There is an expanding body of material on UVLC in which green or blue LEDs or LDs are employed as wireless transmitters [88]. Nevertheless, the majority of these evaluations are restricted to single users and point-to-point linkages. However, the actual application of USNs necessitates the creation of numerous access systems to accommodate multiple sensor nodes. As a result of this, various multiple access strategies for UVLC systems have been developed [89]. The experimental demonstration of NOMA-UVLC with a blue laser transmitter was presented in [90], with a cumulative rate of 4.686 Gbps attained for two users. Some studies investigated the numerical performance of multiuser PD-NOMA over a lognormal fading channel, which is generally applicable in low turbulence situations as experimentally validated in [91]. Ref. [92] examined the error rate performance and feasible capacity of a PD-NOMA-enabled UVLC system over the exponential-generalized gamma (EGG) distribution, which is appropriate for turbulence with air bubbles. To the best of our knowledge, only [82,83] have investigated the PD-NOMA for UVLC over lognormal turbulence channels. The influence of turbulence severity on channel capacity as well as the impact of the number of nodes and targeted rate on coverage probability were explored in [82]. The influence of the PA coefficient on coverage probability and attainable capacity is investigated in [83]. An overview of NOMA-UVLC is illustrated in Figure 5. 

### 3.2. NOMA-Based Hybrid RF/VLC Systems 

Intensity modulation and direct detection (IM/DD) is the mechanism through which VLC is achieved by adjusting the LEDs’ output levels of light. Despite the many potential benefits of VLC systems, their primary downside is a severe drop in performance when a non-line-of-sight (NLOS) element is present. For NLOS wireless communication, that is not the case with traditional RF waves. However, whereas VLC systems are ideal for downlink transmission, they are not suitable for uplink transmission due to the fact that they create undesired irradiance. To address these limitations, hybrid VLC/RF designs have been developed; these systems effectively merge the advantages of RF and VLC communication. Recent years have seen a rise in interest in the cohabitation of indoor VLC and RF due to the possibility it offers of improving communication performance. The main reasons for this are the need for universal service coverage and the demand to get around the shortcomings of VLC in duplex transmission situations. Considering hybrid RF/VLC networks have been proven to significantly improve energy efficiency, range, and total system capacity [93], they have been advocated as a central option for indoor communication networks. On top of that, because the light is commonly restricted inside a relatively small region, such systems offer a solution to problems with undesirable variations of the possible throughput in VLC. Therefore, it is expected that the integration of VLC and RF would considerably enhance the whole user experience. Recent significant contributions have examined several facets of hybrid VLC/RF systems from the perspective of maximizing their combined strengths. In particular, the authors of [94] used an optimal bit-and-power allocation method to study the optimal operation properties of a hybrid VLC-OFDM system. To take advantage of both the high data throughput capabilities of the VLC component and the high connection dependability of the RF component, Wang et al. investigated a hybrid VLC/Wi-Fi system [95]. Based on their investigation of the downlink of a hybrid VLC/RF system, the researchers in [96] demonstrate that the suggested system may greatly improve the total coverage by making use of Wi-Fi in addition to the VLC connection. Likewise, in [97], the capacity performance of a hybrid VLC/RF setup based on OFDM was examined. It is emphasized that in today’s communications, both local and large-scale, the energy economy is of utmost importance. Recent reports on the energy efficiency of hybrid VLC/RF networks have also been highlighted in this area. Equally, the investigators in [98] considered how to allocate energy effectively between single VLC AP and a single RF AP in an OFDMA-aided hybrid VLC/RF system. NOMA, nevertheless, has been offered as a viable alternative to OFDMA for use in future radio access networks.

Downlink data throughput and spectral efficiency in cellular networks may be significantly increased by employing the NOMA approach. NOMA’s defining feature is its ability to multiplex users in the power domain simultaneously, allowing each user to make full use of the available frequency bandwidth. This would be accomplished by the transmitter employing superposition coding and the receiver using SIC to first remove the interference caused by the ensuing information signals and then decode them. In light of these observations, a recent article [99] proposes and analyzes a NOMA-enabled VLC/RF system that, to the researchers’ knowledge, does not exist in the existing literature. Here, the authors measure how much energy it saves during downlink transmission on the network under consideration. To achieve this goal, they first compare the advantages of the studied method to those of its OFDMA counterpart scheme and then derive an analytic expression for the energy efficiency of VLC/RF NOMA. Figure 6 presents an overview of hybrid RF/VLC wireless network. 

An online method was developed by the authors of [100] to reduce the energy used by a hybrid VLC-RF network in a low-light environment while still meeting the need for the available light. Similarly, in [101], the paper analyzed optimum categorization for a NOMA hybrid VLC-RF network in the presence of perfect CSI. To the same end, [102] tackled the issue of efficient resource allocation in NOMA-aided hybrid VLC-RF with shared backhaul to optimize the possible data rate. In order to ensure that vehicular communications are both ultra-reliable and low-latency, the researchers in [103] presented a powerful hybrid VLC/RF paradigm for resource management. In addition, a heterogeneous RF/VLC model for optimizing RF/VLC link selection has been presented [104]. Last but not least, the NOMA-aided hybrid VLC-RF system’s performance was examined in the context of reconfigurable intelligent surface (RIS) by scientists [105].

### 3.3. NOMA-Based VLC System with IRS 

The idea of “intelligent reflecting surfaces” (IRSs) has recently become a hot topic in the wireless communication industry. This is because IRS offers a spectrum-, power-, and cost-efficient way for wireless networks to evolve over the long term. An IRS is made up of a number of reflecting elements (REs) that may be intentionally created to alter the way they react to incident light rays. Based on this, it is feasible to efficiently manage light signal propagation to obtain desired performance benefits. The researchers [106] give a full analysis of the pros and cons of using IRS technology in VLC and LiFi systems. Ongoing studies have concentrated on evaluating and improving VLC performance in IRS-enabled systems. For instance, [107] explored IRS-enabled indoor VLC systems, whereas [108] addressed the optimization of the IRS reflection coefficients with the goal of sum-rate maximization. The authors of [109] describe a framework for an IRS-assisted NOMA-VLC system with the goal of improving connection dependability. The authors look into how adding IRSs to NOMA-based VLC systems could improve their performance. In traditional NOMA-aided VLC, the decoding order of users and, by extension, power allocation may be determined by the LoS channel gain. However, in IRS-aided systems, this is not always the case. As a result, tweaking the IRS allows you to gain control over the apparent overall channel at the receiver. With the goal of improving the BER, the authors present a structure for the coordinated design of the IRS reflection coefficients, NOMA decoding order, and power allocation. In addition, they prove that this multi-dimensional optimization issue is NP-hard and present an adaptive-restart genetic algorithm (GA) to solve it efficiently. The compelling integration of IRS and NOMA has drawn significant research attention in an effort to exploit the new DoFs empowered to NOMA through IRSs [110]. 

In another recent work [111], the authors provide a PLS method for NOMA-VLC systems with IRS support. PLS is shorthand for “physical layer security,” which describes methods that use the inherently unbreakable nature of the optical channel to protect data transmission from snoopers. They propose a challenge to maximize the secrecy capacity of a trusted user by preserving minimal rate limitations for the untrusted user, under the premise that users in the network are given a trust score, i.e., based on their recent behavior. To accomplish this, the optimal PA of NOMA and IRS setup are collectively optimized according to the available system characteristics, user locations, and desired rates. To achieve computational efficiency, the authors offer a novel PLS approach and an alternating optimization technique that employs the adaptive-restart genetic algorithm (GA). IRS-aided NOMA-VLC system for two users is shown in Figure 7. The system is based on a trusted user and an untrusted user. The users can get data from LoS and reflected path from IRS.

### 3.4. NOMA-VLC with UAV 

New research has been done on how combining NOMA and UAV could be a game-changer for 5G and beyond in terms of making connections and coverage available everywhere. In [112], a path-following algorithm was used to tackle a joint optimization issue for NOMA-empowered UAV downlink networks, including bandwidth allocation, antenna beamwidth, UAV altitude, and power allocation. When used to uplink cellular networks, NOMA has been proven to reduce UAV runtime while fulfilling the QoS needs of ground users [113]. In [114], the authors analyze a non-orthogonal multiple access (NOMA) method for uplink UAV-aided wireless communications and compare it to the currently used slotted ALOHA. Both [115,116] took into account the network energy efficiency and sum rate as motivations for jointly optimizing UAV deployment and power distribution. Ref. [117] optimizes user association and UAV placement to reduce total power consumption. Some methods, including machine learning (ML), game theory [118,119], and network optimization [51,112] have been developed to improve UAV/NOMA/VLC systems. In particular, ML has found various uses in 5G wireless networks as a result of the advent of new applications and technology. For instance, [119] considered deep learning as an expanded version of the previous work [117], used federated learning to address a number of issues at the wireless edge, and [120] provided a thorough overview of deep reinforcement learning and its many applications in fields as diverse as wireless caching, edge computing, and network security. Swarm intelligence is seen as a crucial method for improving 5G and beyond networks because of its competitive performance, high dependability, and quick convergence. Since its proposal [121], the Harris hawk optimizer (HHO) has quickly risen to prominence as a cutting edge swarm intelligence approach. 

In order to reduce overall power consumption, [122] developed a scenario in which a UAV’s position and user were linked. The latest research in [123] is the first instance in which the researchers have discussed the topic of employing UAVs to improve NOMA-VLC systems explored in the literature. The authors highlighted in [123] as to why the research community thinks UAV and NOMA integration with VLC is highly promising. To begin with, a NOMA-VLC system with the help of UAVs may supply not only light but also communication services for several users at once, allowing for a vast and pervasive connection for IoT applications in B5G. Furthermore, while sustainable energy and wireless power transfer do help with UAVs’ energy conservation, using UAVs with very low power (VLC) capabilities for communications rather than UAVs with radio frequency (RF) resources is an even better solution. Finally, UAVs’ great mobility and adaptability make it possible to ensure and improve users’ LoS connections and QoS in VLC. The next step is testbed testing, which has been conducted to validate the use of UAVs in VLCs [124]. Simulation findings further show that the suggested method for UAV-assisted NOMA-VLC outperforms OMA and fixed-position (i.e., non-adjustable LED location) schemes.

### 3.5. NOMA-VLC with OFDM 

When it comes to 5G networks, OQAM/OFDM is seen as potential modulation scheme. The use of a filter with a high side-lobe suppression ratio and well-defined time-frequency localization provides OQAM/OFDM with their primary benefits [6]. For this reason, OQAM/OFDM has been shown to be more resilient against ICI in both VLC systems and fiber transmission systems due to its reduced out-of-band power leakage. The spectral efficiency is further enhanced by employing OQAM modulation and a filter bank, both of which eliminate the need for CP. As of now, OQAM and OFDM have been shown to be appropriate for asynchronous carrier amass in the 5G HetNets. However, true orthogonality among subcarriers is still necessary, which in turn restricts the possible number of users. 

OFDM-NOMA in a single-cell VLC system is suggested in [49]. In an earlier work [75], the authors presented OQAM/OFDM-NOMA for a 2 × 2 MIMO-VLC system to increase system capacity. To reduce interference between two neighboring channels, the MIMO equalization approach with unique training sequences (TSs) is utilized. A single-cell VLC system requires spatial combination with two receivers to get the sent signal, and the system performance at a given place is examined. OQAM/OFDM-NOMA modulation is suggested and experimentally proven in a recent study [6] for a multi-user asynchronous multi-cell VLC system. It can increase user fairness among cell-edge and cell-center users inside the same cell and overall system performance as well. PD multiplexing is enabled by NOMA and used by altering the power weight of each user correspondingly. Furthermore, in a multi-cell VLC system, the combination of OQAM/OFDM and NOMA may efficiently decrease intercell interference. The outcomes of the experiments reveal that since the power ratio is 8 dB, OQAM/OFDMNOMA might attain the highly similar BER performance across cell-edge and cell-center users, thereby supporting user fairness.

Recently, experimental and computational studies on the OFDM-NOMA-VLC have been undertaken [53,125]. Ref. [49] studied the OFDM-NOMA-VLC uplink and downlink systems, as well as the effects of channel estimation and power allocation. The performance benefit of non-Hermitian symmetric (NHS) inverse fast Fourier transform (IFFT)/FFT-based OFDM-NOMA-VLC has been demonstrated through an experiment in [125]. The authors of [53] explored an enhanced power allocation (EPA) method for OFDM-NOMA-VLC and showed that it might improve throughput over traditional techniques. Nevertheless, it only addressed power distribution for various users inside each subcarrier, not power allocation across subcarriers. Subcarrier transmission, along with power allocation, is a critical challenge in the actual deployment of the OFDM-NOMA-VLC system. There is currently no requirement for subcarrier allocation in the present OFDM-NOMA-VLC system since it is expected that any subcarrier would multiplex all users. Unfortunately, given the SIC complexity at the receiver, the number of multiplexed users on every subcarrier in a realistic OFDM-NOMA system should be restricted. The proportion of multiplexed users for each subcarrier in OFDM-NOMA could be smaller than the overall number of users, but subcarrier allocation can still maintain the user fairness. Ref. [126] designed a software-based NOMA-VLC system featuring dynamic power and carrier allocation, but somehow it did not investigate the particular subcarrier allocation technique. To facilitate large-scale device connections and reduce energy, [89] suggested a power allocation and subcarrier technique for asymmetric clipped optical (ACO) OFDM-NOMA for an uplink underwater VLC system. This approach, unfortunately, cannot be used for DC-biased OFDM-NOMA in indoor VLC systems because the electrical signal’s peak amplitude must be controlled to assure human eye safety and optical signal non-negativity. A generic overview of two-user NOMA-based VLC system with QAM and O-OFDM is presented in Figure 8. 

In Figure 9, we have provided BER performance analysis for NOMA-DCO-OFDM for two users (User1 and User2). We analyzed the impact of power allocation on BER performance for each user. In Figure 9a, we consider power allocation as 0.9 and 0.1 for User1 and User2, respectively. While in Figure 9b, we consider power allocation as 0.6 and 0.4 for User1 and User2, respectively. Our results show that a significant power allocation difference between both users can give better BER performance.

### 3.6. Machine Learning Techniques for NOMA-VLC

With the introduction of machine learning (ML) algorithms, particularly deep learning (DL) and neural networks (NNs), VLC has the potential to become more efficient and solve numerous issues at the physical layer [127]. Multi-CAP and Nyquist pulse amplitude modulation (PAM) VLC systems were addressed in [128,129], respectively, with a clustering algorithm-aided perceptual choice approach and a nonlinear compensation mechanism. In order to compensate for PAM-transmission VLC’s flaws, a post-equalizer based on deep long-short-term memory (LSTM) was developed. The use of red, greenish, and blue LEDs in deep learning (DL)-aided multi-colored VLC connection was described in [130]. With the use of machine learning, the authors of [131] explored the possibility of using this technique to build and implement a VLC connection for demodulating signals.

To address the linear and nonlinear deformities that plague NOMA-VLC transmissions, the authors in [127] present a convolutional NN (CNN)-based signal demodulator as presented in Figure 10. Recorded NOMA signals are utilized for: (i) offline CNN training and (ii) online signal compensation and recovery using a CNN-based demodulator. Remember that the NOMA signal can be detected without the free-space channel response. They demonstrate, via modeling and experimental data, that the recommended CNN-based demodulator would successfully attenuate nonlinear and linear distortions, therefore enhancing the system’s functionality.

To increase the spectral efficiency of a MIMO-NOMA system when facing interference from a smart jammer, the researchers in [132] suggested a swift RL-based power allocation technique. For mobile edge computing with NOMA, the research in [133] employed Q-learning to create a framework. DRL solves a problem with Q-learning related to Q-table storage and lookup by combining deep learning into RL. Power allocation in cache-assisted NOMA systems was created by [134] authors using dynamic resource balancing. A DRL was employed in [135] to find sub-optimal power allocation strategies for an uplink multicarrier NOMA system. In conclusion, a combined channel assignment and power allocation problem in a two-user NOMA system was addressed by He et al. [136] utilizing a DRL framework. The key goal [137] is to find the configuration between optimal power distribution and optimal LED transmission angle adjustment for a virtual local area network (VLAN) with users spread out evenly around the space. The issue is NP-hard, meaning it cannot be solved by traditional optimization techniques in this context. DQL’s greatest strength is in its ability to address difficult joint optimization issues in wireless communication, issues that are typically intractable with the use of more traditional mathematical analysis [138]. Several recent publications have shown that DQL is effective. For example, the researchers in [139] used DQL to forecast and modify the IRS phase shift matrices optimally, allowing for the optimization of an IRS-NOMA system. Although it is impossible to get complete information about the channel state, the DQL method enables the agent to acquire new knowledge about the nature of communication, which can then be used to find the best possible solution. In [123], the authors proposed an energy efficient HO algorithm for UAV-aided VLC-NOMA. As a swarm intelligence tool, the Harris hawks optimizer (HHO) is among the most contemporary algorithms that have gained popularity since its inception. It formulated a joint problem by considering UAV’s placement and PA to maximize the sum rate of all users.

### 3.7. Physical Layer Security (PLS) in NOMA-VLC 

Recent research on PLS in NOMA networks released after the 2016 study by Zhang et al. [140] has been on NOMA users who are either snooped on or apprehended by exterior hostile snoopers [141]. Researchers have also looked into how certain users behave as fake nodes in comparison to other, more reputable NOMA users [142]. In order to create a foolproof method of transmission, the researchers behind [143] took into account the possibility of both internal and external eavesdropping. Analysis of NOMA system secrecy performance is possible thanks to the accepted mathematical tools of game theory [144], stochastic geometry theory [141], and optimization theory [140]. With the right signal processing, PLS for NOMA systems may use the same transmit antenna selection [145] that is used in PLS for VLC networks. Although PLS in VLC systems is only relevant to downlink wireless communication, in NOMA systems it may also be used for uplink transmission [146]. Nevertheless, due to the unique qualities of the optical wireless transmission channel and optical transceiver, these studies on PLS in NOMA systems are limited to the RF domain and it is difficult to directly extend to the VLC domain. Research on PLS in NOMA-aided VLC networks has been done on many occasions. Secrecy outage probability (SOP) for a multiuser, multi-external-eavesdropper downlink NOMA-VLC network was inferred in [147], which showed that SOP performance was linked to the variables of the snooping density and optical transceiver, according to the spatial distribution of eavesdroppers and trusted users. Numerous recognized relays with the optical transceiver have been developed, and secure beamforming vectors have been tailored [148] to guarantee safe transmission in a two-user, single-external-eavesdropper downlink NOMA-VLC network, demonstrating that the ideal relaying technique changes depending on the number of relays and the geometric configuration. Until now, however, only genuine users in a static state have been included in PLS studies in NOMA-enabled VLC networks. In [149], the authors transform a resource allocation problem into the problem of dynamically allocating power to address PLS for mobile users in NOMA-VLC systems. To the experts’ knowledge, however, research into AN-assisted secure beamforming for a VLC-NOMA system has not yet been published. To help bridge that gap, a recent work [150] investigates the secure resource allocation issue for VLC-NOMA systems. In the case of NOMA in particular, it is important to keep in mind that the performance increases are heavily dependent on the precision of the CSI. Therefore, a realistic, inaccurate CSI of both the targeted users and eavesdroppers is assumed in order to properly examine the performance increase realized by an AN-assisted stable beamforming structure in a multiple-input, single-output (MISO) NOMA-VLC system. Particularly, the inter-user interference issue in NOMA-VLC systems can be efficiently mitigated through exploiting the spatial DoFs to design appropriate transmitter and receiver beamformers. Figure 11 presents an overview of a NOMA-VLC system with two trusted users and an eavesdropper who can steal data. 

## 4. Potential Challenges and Open Research Issues

NOMA may be used to solve a variety of problems in VLC systems as shown in Figure 12. However, implementing NOMA in VLC systems introduces new limits and obstacles that necessitate appropriate solutions. Based on the existing work in the literature, in this section, we outline various challenges and open research issues that need to be considered and investigated in the future work.

### 4.1. MIMO

Indoors, enough lighting is typically provided by arrays of LEDs, which drive the need for MIMO in VLC systems. When MIMO methods are used in VLC systems, the limited modulation bandwidth constraint of the LEDs may be circumvented, leading to improved spectral efficiency [151]. Using spatial multiplexing, users may be divided into groups, with each group being assigned to a different light-emitting diode (LED). As in the power domain, users from the same group that share the same modulation bandwidth are overlaid. Multiple users of MIMO-VLC systems employ precoding methods to partition incoming signals into their respective MIMO subchannels [76]. As an added bonus, SIC may be used to prevent intra-group interference among multiplexed users. Exploiting MIMO-VLC has the potential to greatly increase the system’s capacity by catering to a significant number of clients all at once [152]. However, the construction complexity of MIMO-PD-NOMA-aided VLC systems is increased because PD-performance NOMA is highly dependent on the power allocation approach and the user grouping approach was chosen. Generally, system performance improves as the rank of the MIMO channel matrix grows because more de-correlated channels are produced. Because of this, the rank of the channel matrix is crucial for judging the performance of MIMO-NOMA. On the contrary, existing research in MIMO-NOMA takes into account full-rank channel matrices to study the performance of the system. 

### 4.2. Security

As opposed to RF channels, VLC channels have their own unique traits. Since light cannot pass through walls, its wireless transmission is safe from any outside eavesdroppers in a VLC network, making it a more secure alternative to RF networks. VLC networks must be secured because of their broadcast nature, making signals vulnerable to eavesdropping in public places like libraries and shopping malls, as well as in multi-user indoor environments like conference rooms. Exploration at the physical level is often linked to ensuring reliable signal transmission, whereas the upper layers are typically accountable for protecting the connection by leveraging encryption technology. Moreover, a potential complementary strategy is offered to improve the protection of wireless communication by means of the physical layer [153]. This is because of the necessity for larger data speeds. An alternative to encryption that takes advantage of the properties of the wireless medium is physical layer security (PLS) [154]. By identifying the difference in SNR between the legitimate user and the eavesdropper, PLS mitigates the negative effects of the wireless channel, such as noise, fading, and the use of multiple transmitters (such as cooperative relays or multi-LEDs), thereby allowing for a more secure transmission. PLS is mostly needed to enhance network information security. Because of the channel capacity mismatch between both platforms, the results of prior research on PLS in RF cannot be directly transferred to VLC, where the critical challenge of nonlinear distortion on VLC systems and positive real-valued signal are needed. Recently, there has been a lot of interest in PLS research for NOMA-VLC networks employing various strategies, such as artificial noise (AN) and beamforming [142]. Aside from PLS, the inventors of [155] used a two-level chaotic encryption approach in the PD-NOMA-VLC system to guarantee both privacy (among legitimate users) and security (against eavesdroppers).

### 4.3. Hybrid VLC/RF Systems

About 80% of all wireless data traffic is thought to be generated by indoor applications. Therefore, we envision VLC as a complementary technology that provides various benefits over RF, particularly in indoor situations, such as utilizing the existing lighting infrastructure to construct a low-cost system that enables ultra-high data rate [156]. While VLC has several distinct benefits, it also has a few downsides that might greatly hinder the efficiency of the VLC system if they are not accounted for. In contrast to RF, which may make good use of the possibilities presented by NLOS communication, the primary difficulty with VLC is the considerable performance decrease that occurs when LOS connectivity is unavailable. The uplink transmission situation in VLC systems is inefficient because of the undesired irradiance, unlike the downlink transmission situation. In this regard, creating a hybrid VLC/RF heterogeneous network (HetNet) that could integrate the benefits of both systems to boost users’ mobility and offer ubiquitous coverage [157] is a potential route to solve these shortcomings. Hybrid RF/VLC networks have gained a lot of interest as of late due to the fact that they are a practical option that can boost indoor communication efficiency. Furthermore, NOMA can improve connectivity and spectral efficiency in these hybrid networks.

### 4.4. Impact of Transmission Distortion 

It is generally accepted that it is lossy to send raw data like audio and video over the internet. At some point in their journey to the receiver, all transmissions suffer distortion. There has been a lot of theoretical work done on measuring source fidelity over fading channels as a means of coping with the lossy transmission. Distinct forms of channel coding and source coding have been designed to lessen the overall amount of distortion introduced during transmission. However, preferences for deformation, cost, and complexity might be at odds with one another depending on whether one uses source or channel coding diversity. The likelihood of an outage has an effect on both information capacity and distortion. Clearly, the maximum outage rate associated with a given outage probability may not result in the least amount of predicted distortion [158]. The greatest outage rate that can be achieved with tolerable distortion using a NOMA system can be investigated by conducting an inquiry to optimize the outage probability.

### 4.5. Impact of Interference 

Some studies concentrate on cooperative NOMA [36] and employs Bluetooth-like short-range communications in the cooperative phase, despite the fact that interference analysis is a generic term in wireless networks. Nevertheless, present wireless personal area network (WPAN) operations provide a severe interference problem for the usage of BT radio in cellular networks. Interference from other BT devices reduces coverage and throughput, disrupts connectivity temporarily or permanently, and makes pairing with other devices challenging for the user during the discovery process. The deployment of channel allocation is influenced by factors such as the amount of interference present in the operating environment, the size of the payload, and the distance between cooperating users. Since NOMA pairs users based on their CSI, a self-organizing scatternet for BT node management needs to be modified so that it may be used in conjunction with NOMA. Even when users’ mobility results in the full loss of some of the wireless links, a resilient scatternet should nevertheless provide viable routes between nodes with a high probability. The movability of users also introduces a new level of complexity to the interference problem. Therefore, it will be interesting to analyze the performance of a cooperative NOMA strategy in this dynamically interfering scenario.

Recently, rate-splitting multiple access (RSMA) has emerged as a promising PHY-layer paradigm for interference management, non-orthogonal transmission, and multi-user communication. RSMA is a universal MA scheme that subsumes OMA, NOMA, SDMA, and multicasting as sub-schemes. For instance, considering multiple-antenna settings, PD-NOMA becomes sub-optimal compared to RSMA [159]. In short, RSMA is a superset of current MA techniques and therefore always offers equal or better performance. Through exploiting the splitting of user messages along with non-orthogonal transmission of common messages decoded by multiple users and private messages decoded by their corresponding users, RSMA shows the potential to reconcile the two extreme interference management techniques of treating interference as noise and fully decoding interference used in NOMA and SDMA [159]. The flexible framework of RSMA ensures better performance for all levels of interference. In case of strong or weak interference, RSMA automatically reduces to NOMA or SDMA through tuning powers and contents of users’ data streams. 

### 4.6. Practical Channel 

The availability of spectrum and a highly efficient radio access technology are two essential components of next generation wireless networks that must be met in order to handle the ever-increasing amounts of user data. It is currently evident that 5G will take advantage of spectrum allocations in the underutilized millimeter wave (mmWave) frequency ranges. In addition, 5G’s backbone networks are anticipated to switch from fiber and copper to mmWave wireless connections, which would enable speedy rollout and a network topology similar to a mesh. The mmWave frequencies between 30–300 GHz represent a new, bandwidth-rich prospect for mobile networks. Therefore, it is crucial to learn about the difficulties of mmWave cellular communications and the channel behavior in particular while designing 5G mobile systems and backhaul methods [160]. Current NOMA research often assumes that the wireless connections between the transceivers display a Rayleigh fading channel with AWGN. If the observed delay spread and route loss values can be taken into consideration to reflect the radio channel in the mmWave band, a more accurate picture of the channel’s behavior would emerge. Similarly, researchers should focus on challenges of mmWave NOMA systems [161]. 

### 4.7. Decoding Complexity 

Due to the fact that the receiver must decode the information of other users before it can decode its own information, implementing SIC-based signal decoding is more difficult than orthogonal systems. More users in the targeted cell mean a higher complexity level. Users, though, can be sorted into groups, with a few poor channels in each category. Then, inside each group, SC/SIC procedures can be carried out. There is a tradeoff between implementation complexity and performance improvements offered by this group-wise SC and SIC operation. In addition, SIC introduces extra delay and complexity, which can increase with the number of users. Several works have addressed these issues by replacing SIC with low-complexity treating interference as noise decoding, i.e., single-user decoding [162]. 

### 4.8. Signaling and Processing Overhead 

When compared to its orthogonal variants, NOMA incurs a higher cost from a variety of signaling and processing sources. Specific time intervals, for instance, must pass in order to gather the CSI from various receivers and update the receivers of the SIC order. Because of this, NOMA’s data transfer rates degrade. NOMA signal processing also has a higher energy need because of dynamic power allocation and SC/SIC encoding/decoding.

### 4.9. Limited Number of User Pairs 

The advantages of NOMA are made possible by its use of power-domain multiplexing. In order to pair a user at the cell’s periphery with a user in the cell’s center, there must be a propagation loss difference of roughly 8 dB. As a result, the capacity benefit of NOMA is diminished since the number of user pairs is capped in a typical NOMA design. It is crucial to find a method of signal recognition and decoding that can boost the capacity for user pairs [163].

### 4.10. Power Allocation Complexity 

The amount of transmit power provided to a user influences their throughput. Because power-domain user multiplexing is the foundation of NOMA, the power allocation also impacts the possible capacity of other users. If we want the highest potential throughput in NOMA, we need to resort to a brute-force search of all possible user pairings with dynamic power allocation. Unfortunately, performing such a thorough search takes a lot of processing power. Thus, future research works will need to pay attention to this research area. 

## 5. Future Research Directions 

Based on our learning from the existing work in the literature, in this section, we explore various research aspects and gaps that need to be considered and investigated in the future work. The following discussion focuses on some of the most salient research difficulties and obstacles that have been found for NOMA-aided VLC systems. As an example, cell-edge users, who are typically seen as weak users, have been given priority [164]. Inter-cellular interference might potentially rise as a result of this. Moreover, in multi-user NOMA-aided VLC systems, the SIC induces significant delay and computational complexity at the receiver end to allow the huge connection. Keeping latency low while increasing the decoding order is a formidable problem. It is difficult to keep track of extensive lookup tables for a high number of users in multi-user multi-cell networks, as described in [165], while utilizing the S-GRPA approach for power allocation. Keeping user locations static or stable in indoor NOMA-aided VLC systems is a requirement of the user-pairing strategy; this is problematic in a real-world setting and calls for further study into how to locate users in a time-sensitive manner. In addition, the SIC decoding order rises in tandem with the user count as it continues to rise. The propagation of mistakes increases with the SIC decoding order, and it is difficult to keep them to a minimum. Security and privacy concerns in NOMA-aided VLC systems for a two-user situation have been addressed in [5]. Each user’s safety and privacy may be ensured in multi-cell, multi-user indoor VLC networks. Maximizing the sum rate using UPPA techniques in NOMA-based VLC systems presents two significant challenges: quality of service and user fairness. Throughout the power domain, NOMA always favors weak users over strong, hence the strong users’ quality of service (QoS) suffers. As a result, these less capable individuals often end up being the ones that produce imbalances of power within the user base. Given that it is nearly impossible to achieve in real time indoor NOMA-VLC systems [166], the assumption of uniform channel gain disparity between a pair of users severely limits the applicability of the user-pairing methods examined there. Although outage probability and sum rate analyses of NOMA-aided VLC systems receive the lion’s share of attention in the research, BER performance may be considered alongside these metrics. 

## 6. Conclusions

This article surveys the existing studies and research contributions of NOMA-based VLC systems. We provide a comprehensive review of NOMA schemes for VLC systems, which contains an extensive coverage of the existing literature. This study includes the integration of NOMA into VLC systems and indicates how it enhances the overall communication performance of VLC systems. We have shown, without a doubt, that the implementation of NOMA techniques in VLC is paramount to minimize the interference and enhance the performance of VLC systems. In this context, we provide detailed discussion about related works through existing literature. This article comprehensively discusses the extensive scope of NOMA-based VLC systems with the integration of several emerging technologies such as IRS, OFDM, UAV, ML, PLS, MIMO, etc. In addition, we briefly explain the benefits and shortcomings of incorporating NOMA-based hybrid RF/VLC systems. Furthermore, we highlight and summarize the potential challenges and open research issues. Finally, a number of future research directions associated with NOMA-based VLC systems have been presented for ameliorating the performance of the NOMA techniques into VLC systems. We firmly believe that this article will capture the interest of both academic and industrial fraternities to investigate the potential of NOMA-based VLC systems in next generation wireless communication networks.

## Figures and Tables

**Figure 1 sensors-23-02960-f001:**
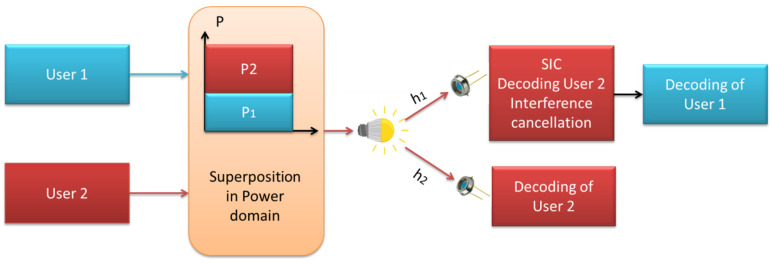
An overview of downlink NOMA-VLC.

**Figure 2 sensors-23-02960-f002:**
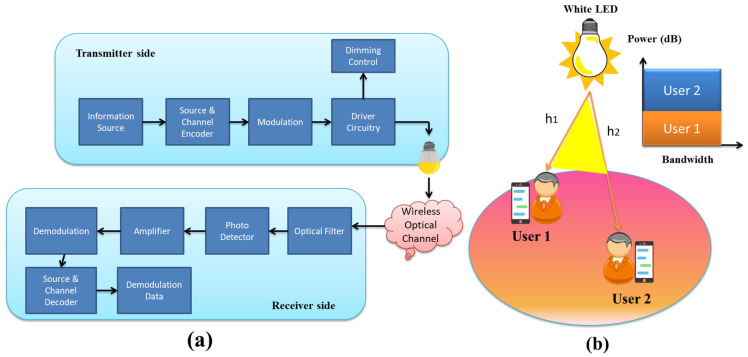
(**a**) An overview of a basic VLC system. (**b**) Conceptual diagram of a NOMA-VLC system.

**Figure 3 sensors-23-02960-f003:**
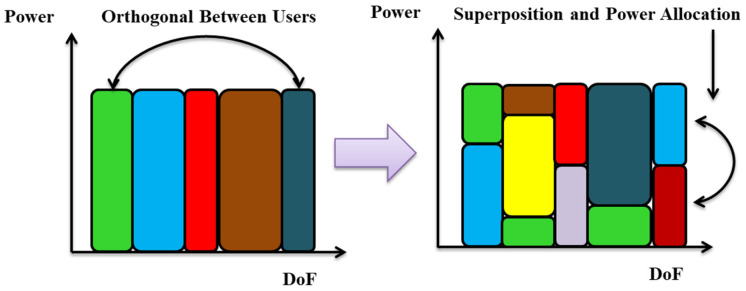
From OMA to NOMA through power domain multiplexing.

**Figure 4 sensors-23-02960-f004:**
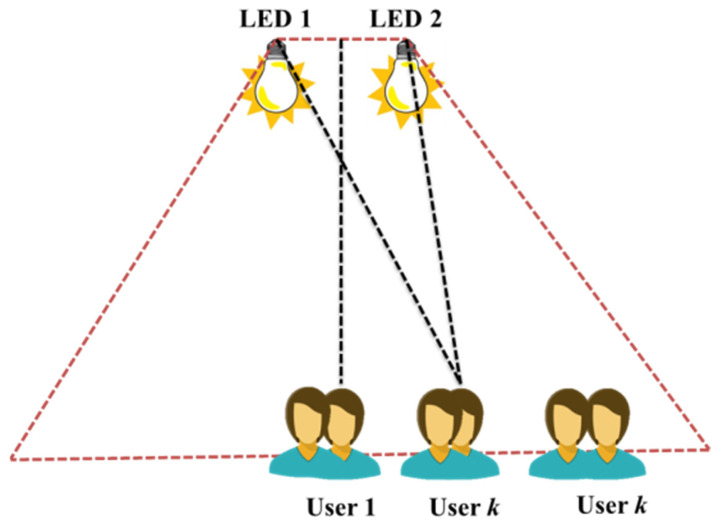
An overview of 2 × 2 MIMO-NOMA-based VLC system with *k*-users.

**Figure 5 sensors-23-02960-f005:**
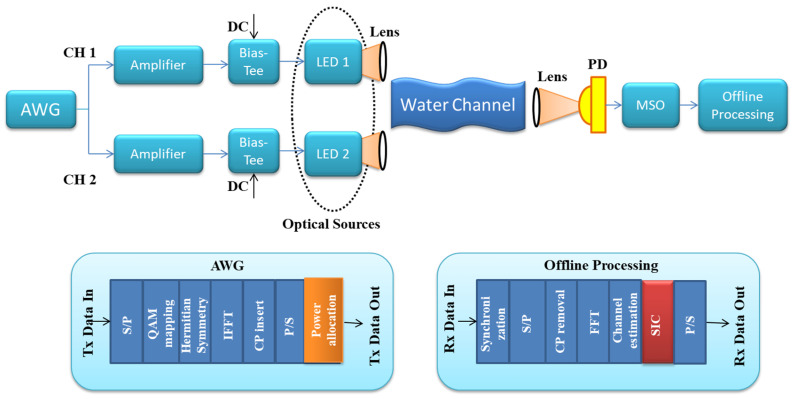
An overview of NOMA-UVLC system.

**Figure 6 sensors-23-02960-f006:**
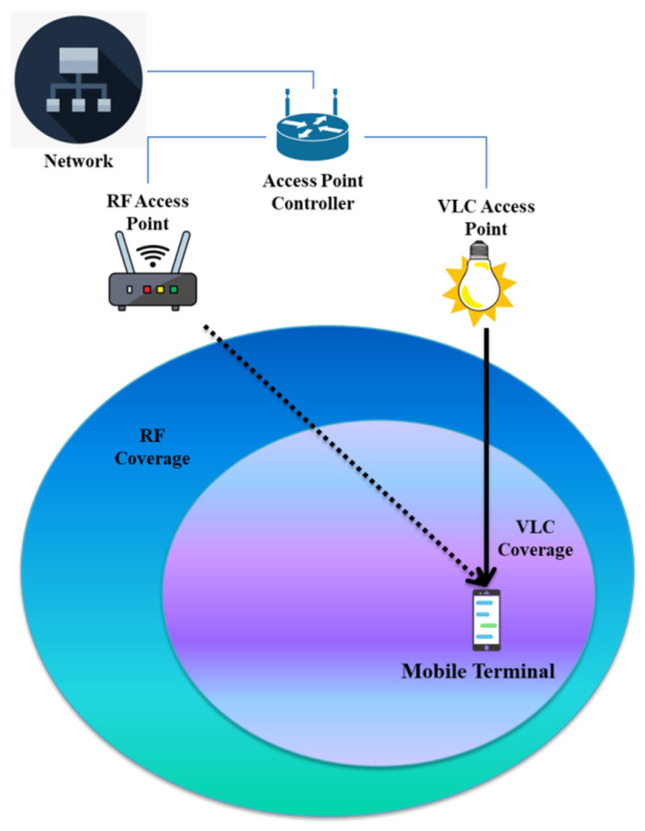
An overview of Hybrid RF/VLC wireless network.

**Figure 7 sensors-23-02960-f007:**
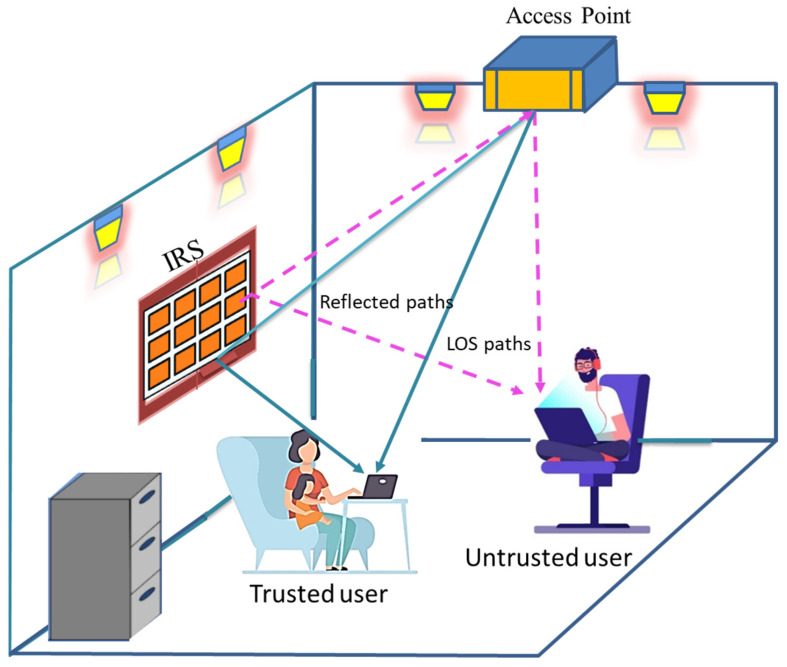
System model with two users: trusted and untrusted.

**Figure 8 sensors-23-02960-f008:**
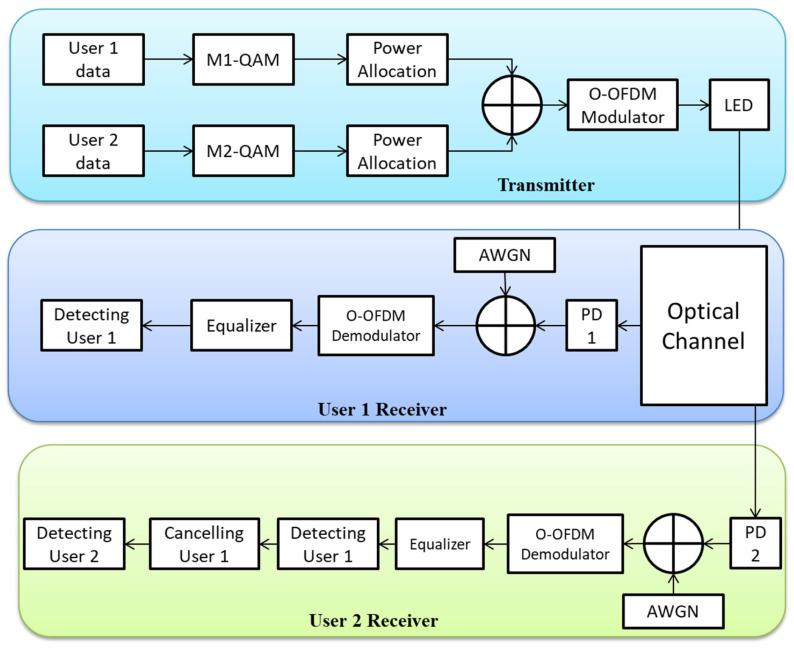
Block diagram of two-user NOMA-based VLC system with QAM and O-OFDM.

**Figure 9 sensors-23-02960-f009:**
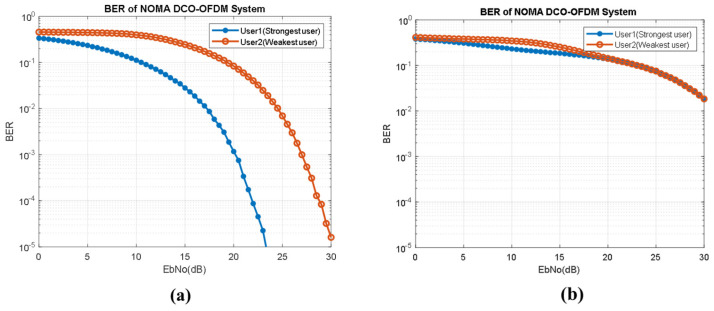
BER performance analysis for NOMA-DCO-OFDM (**a**) User1 PA 0.9 and User2 PA 0.1 (**b**) User1 PA 0.6 and User2 PA 0.4.

**Figure 10 sensors-23-02960-f010:**
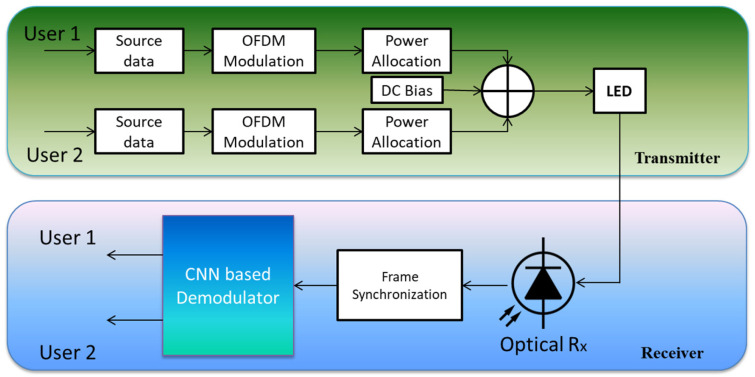
An overview of NOMA-VLC system using a CNN-based demodulator.

**Figure 11 sensors-23-02960-f011:**
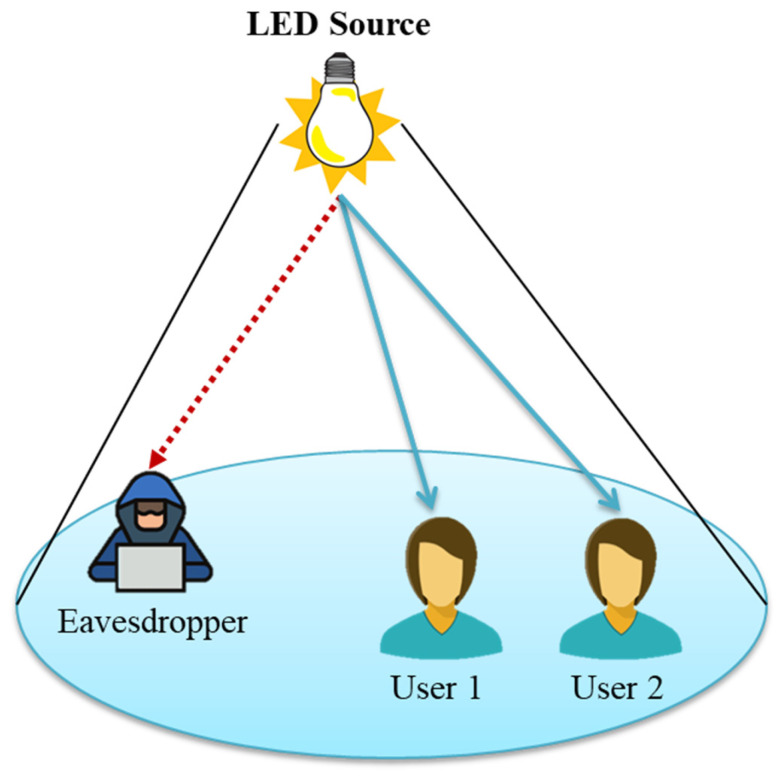
NOMA-VLC system with two users and an eavesdropper.

**Figure 12 sensors-23-02960-f012:**
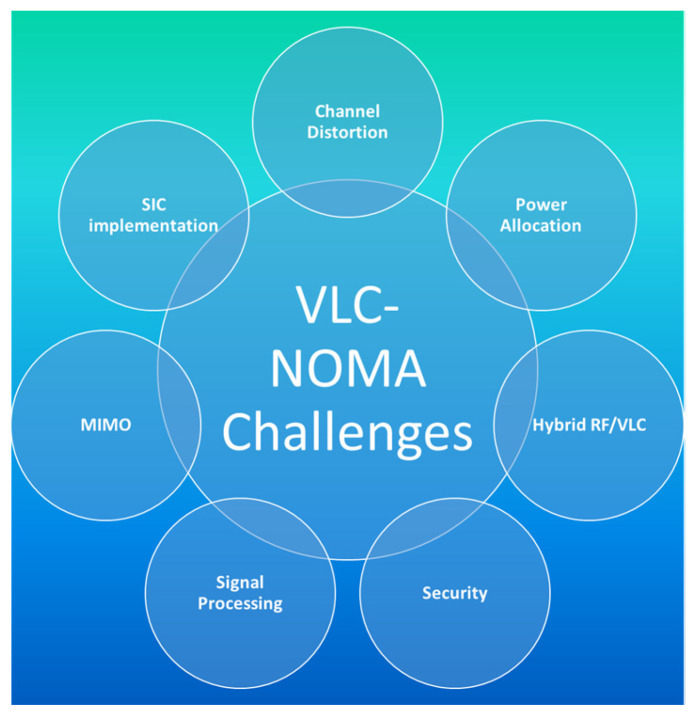
Critical challenges in NOMA-VLC.

**Table 1 sensors-23-02960-t001:** Summary of recent works [14].

Reference	Communication Domain	Research Contribution
[8]	5G networks	Review of challenges and potentials of PD-NOMA for 5G systems
[15]	5G networks	It surveys NOMA techniques for 5G networks
[16]	6G networks	It provides future vision and research opportunities for next generation NOMA
[5]	VLC	GRPA scheme to enhance performance of NOMA systems
[14]	VLC	EPA method to enhance energy efficiency of NOMA-aided IoT sensor networks
[17]	VLC	NGDPA scheme to improve capacity of NIMO-NOMA
[18]	VLC	Power allocation and user pairing methods for downlink NOMA-VLC
[19]	VLC	Survey of research challenges and future trends for VLC-NOMA
[20]	RF communication	DDPA technique for mMIMO-aided NOMA
[21]	RF communication	Energy-efficient PA for multiuser MIMO-NOMA
[22]	RF communication	PA to ensure QoS requirements in NOMA systems
[23]	RF communication	PA to ensure individual QoS requirements in downlink NOMA systems
[24]	Hybrid RF/VLC	Link selection and user pairing in Co-NOMA systems
[25]	Hybrid RF/VLC	Improvement of reliability and outage performance of co-NOMA systems

**Table 2 sensors-23-02960-t002:** A comparison between the MA techniques in successive generations of cellular networks.

Characteristics	1G	2G	3G	4G	5G
Time span	1970–1980	1990–2004	2004–2010	2010–Now	Around 2020
MA technique	FDMA	CDMA/TDMA	CDMA	OFDMA	NOMA
Physical resource	Frequency	Time	Time/PN codes	Orthogonal frequency	Power domain/Code domain
Network’s core	PSTN	PSTN	Packet network	Internet	Internet
Duplex mode	FDD	FDD	FDD/TDD	FDD/TDD	FDD/TDD
Technologies	NMT, AMPS	IS-54, GSM	EDGE, UMTS	LTE, LTE-A, Wimax	Mm Waves, MIMO
Frequency	30 kHz	1.8 GHz	1.6–2 GHz	2–8 GHz	3–30 GHz
Data rate	2 Kbps	64 Kbps	2 Mbps	1 Gbps	>1 Gbps
Hand off	Horizontal	Horizontal	Horizontal	Horizontal/Vertical	Horizontal/Vertical
Services	Analog voice	Digital voice, SMS, MMS	Audio/Video	Mobile multimedia, wearable devices	IoT, video streaming, interactive multimedia, 3D games

**Table 3 sensors-23-02960-t003:** Summary of recent works on NOMA-based VLC.

Reference	Objective	Research Findings
[6]	Evaluation of error vector magnitude, BER, and spectral efficiency	Given technique is more robust and outperforms OFDM-based NOMA
[17]	To maximize the sum rate	NGDPA enhances the sum rate performance as compared to GRPA
[34]	Ergodic sum rate and coverage probability analysis	NOMA outperforms traditional OMA technique
[53]	To maximize the sum rate	The performance of NOMA-OFDM is better than OMA-OFDM in the context of achievable data rate
[56]	BER analysis	Closed-loop expressions for BER validate simulation results
[58]	Rate splitting	Offers an overview of MA techniques in VLC systems. It proposes rate-splitting multiple access (RSMA) and highlights its potentials and capabilities in VLC systems.
[59]	To maximize the sum rate	WDM-NOMA outperforms NOMA in the context of sum rate
[62]	Outage probability analysis	Rate splitting trade-off permits outage performance balancing among users
[63]	To maximize the sum rate	Game theory based optimal power allocation and user grouping
[64]	Evaluation of user fairness, outage probability, and sum rate	Dynamically choosing the appropriate MA technique that attains better performance
[65]	To maximize the sum rate and max-min rate criteria	Optimized power allocation and user grouping to achieve high sum rate than OMA
[66]	BER analysis	Enhanced BER performance compared to NOMA considering SIC for various power levels
[67]	SER analysis	Users at various locations attain identical SER through adequate power allocation

## Data Availability

Not applicable.

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
