# Peer review of "NOMA-Based VLC Systems: A Comprehensive Review"

_sensors, 2023, doi:10.3390/s23062960_

Round 1

Reviewer 1 Report

1. A handful of statements are not precise. For example, on p2 “while it uses multi-user detection (MUD) at the receiver, including successive interference cancellation (SIC)”. Note that SIC is decoding rather than detection. The reviewer suggests that the authors not to mix detection with decoding.

2. SIC introduces extra delay and complexity, which can increase with the number of users. Several works have addressed these issues by replacing SIC with low-complexity treating interference as noise decoding, i.e., single-user decoding. The reviewer would like to suggest the authors to include these important works, e.g., [R1], in this survey paper.

[R1] M. Qiu, Y. -C. Huang, S. -L. Shieh and J. Yuan, "A Lattice-Partition Framework of Downlink Non-Orthogonal Multiple Access Without SIC," in IEEE Transactions on Communications, vol. 66, no. 6, pp. 2532-2546, June 2018, doi: 10.1109/TCOMM.2018.2805895.

3. The paper cites some references on rate splitting multiple access (RSMA). However, a detailed introduction of this subject is not provided. In addition, RSMA includes PD-NOMA as a special case [R2]. This should be mentioned clearly in the paper. Further, in multiple-antenna settings, PD-NOMA becomes sub-optimal compared to RSMA [R2].

[R2] Y. Mao, O. Dizdar, B. Clerckx, R. Schober, P. Popovski and H. V. Poor, "Rate-Splitting Multiple Access: Fundamentals, Survey, and Future Research Trends," in IEEE Communications Surveys & Tutorials, vol. 24, no. 4, pp. 2073-2126, Fourthquarter 2022, doi: 10.1109/COMST.2022.3191937.

4. Please emphasize the key differences between VLC-NOMA and RF-NOMA. The current paper reads like direct applications of many techniques from RF-NOMA to VLC-NOMA.

5. The presentation is very high level. It is better to include the description of the detailed schemes and research insights.

Author Response

Dear Reviewer,

Thank you very much for your valuable remarks which greatly helped us to improve the quality of our work. Please find our response letter in the attached document.

Reviewer 2 Report

This paper is in general well written. However, there exists the following problems:

1) You claim "Yet, despite their appealing capabilities, VLC systems face several limitations which constraint their potentials." However, what are the specific challenges? Please highlight them.

2) As compared with OMA, it should mention the concept of DoF, please comment on the following paper: On DoF of Active RIS-Assisted MIMO Interference Channel with Arbitrary Antenna Configurations: When Will RIS Help?

3) Could you add some simulation results to support your viewpoints?

Author Response

(The authors gave the same response as above.)

Round 2

Reviewer 1 Report

My comments have been addressed. I do not have any further comments.

Reviewer 2 Report

This revision is pretty good.